# New Paradigms of Old Psychedelics in Schizophrenia

**DOI:** 10.3390/ph15050640

**Published:** 2022-05-23

**Authors:** Danish Mahmood, Sattam K. Alenezi, Md. Jamir Anwar, Faizul Azam, Kamal A. Qureshi, Mariusz Jaremko

**Affiliations:** 1Department of Pharmacology & Toxicology, Unaizah College of Pharmacy, Qassim University, Unaizah 51911, Saudi Arabia; sk.alenezi@qu.edu.sa (S.K.A.); m.anwar@qu.edu.sa (M.J.A.); 2Department of Pharmaceutical Chemistry & Pharmacognosy, Unaizah College of Pharmacy, Qassim University, Unaizah 51911, Saudi Arabia; f.azam@qu.edu.sa; 3Department of Pharmaceutics, Unaizah College of Pharmacy, Qassim University, Unaizah 51911, Saudi Arabia; ka.qurishe@qu.edu.sa; 4Smart-Health Initiative (SHI) and Red Sea Research Center (RSRC), Division of Biological and Environmental Sciences and Engineering (BESE), King Abdullah University of Science and Technology (KAUST), Thuwal 23955, Saudi Arabia; mariusz.jaremko@kaust.edu.sa

**Keywords:** psychedelics, schizophrenia, LSD-like drugs, serotonin, mGl2/3 receptors, glutamate

## Abstract

Psychedelics such as lysergic acid diethylamide (LSD), psilocybin (magic mushrooms), and mescaline exhibit intense effects on the human brain and behaviour. In recent years, there has been a surge in studies investigating these drugs because clinical studies have shown that these once banned drugs are well tolerated and efficacious in medically supervised low doses called microdosing. Psychedelics have demonstrated efficacy in treating neuropsychiatric maladies such as difficult to treat anxiety, depression, mood disorders, obsessive compulsive disorders, suicidal ideation, posttraumatic stress disorder, and also in treating substance use disorders. The primary mode of action of psychedelics is activation of serotonin 5-HT_2A_ receptors affecting cognition and brain connectivity through the modulation of several downstream signalling pathways via complex molecular mechanisms. Some atypical antipsychotic drugs (APDs) primarily exhibit pharmacological actions through 5-HT_2A_ receptors, which are also the target of psychedelic drugs. Psychedelic drugs including the newer second generation along with the glutamatergic APDs are thought to mediate pharmacological actions through a common pathway, i.e., a complex serotonin–glutamate receptor interaction in cortical neurons of pyramidal origin. Furthermore, psychedelic drugs have been reported to act via a complex interplay between 5HT_2A_, mGlu2/3, and NMDA receptors to mediate neurobehavioral and pharmacological actions. Findings from recent studies have suggested that serotoninergic and glutamatergic neurotransmissions are very closely connected in producing pharmacological responses to psychedelics and antipsychotic medication. Emerging hypotheses suggest that psychedelics work through brain resetting mechanisms. Hence, there is a need to dig deeply into psychedelic neurobiology to uncover how psychedelics could best be used as scientific tools to benefit psychiatric disorders including schizophrenia.

## 1. Introduction

Schizophrenia remains a serious chronic mental illness [1] since its revelation more than a century ago by Dr. Emile Kraepelin. Despite the low prevalence, nearly 24 million people suffer from this disorder, which constitutes 1 in 300 people (0.32%) of the world’s population and this rate is 1 in 222 people (0.45%) among adults [2,3]. The symptoms of schizophrenia more often appear in the second or third decade of life, and disease occurrence is tied to a combination of factors such as genetic, socio-demographic, and environmental factors [4]. Clinical schizophrenia is presented in two unique and distinct sets of symptomatology, which include ‘positive’ symptoms and ‘negative’ symptoms, and is also accompanied by significant impairment of cognitive functioning in one or more major areas. This may include an inability to execute work, interpersonal relations, or self-care, and there is also a failure to achieve the expected level of interpersonal, academic, or occupational functioning [5,6]. According to the current Diagnostic and Statistical Manual for mental disorders-V (DSM-V), the positive symptoms of schizophrenia are delusions, hallucinations, disorganized speech, and behaviour; and the negative symptoms are diminished emotional expression or avolition [5]. These symptoms have been found to be chronically present once the disease starts, but generally the illness is marked as alternate signs of remission and exacerbation or partial remission or exacerbation. Some psychotic symptoms may be treated without the need for medication with proper human care, social support and care including electroconvulsive therapy [7]. However, pharmacotherapy remains the cornerstone to control psychotic symptoms and prevent recurrence.

### 1.1. Therapeutic Armamentarium for Schizophrenia

The first effective medication for treating psychotic symptoms was reserpine: an antihypertensive agent whose efficacy in schizophrenia was correlated to the reduction in synaptic dopamine release. The discovery of reserpine paved the way for the first antipsychotic medication, chlorpromazine in 1952 [8,9], which was originally synthesised as an anaesthetic agent [10]. The efficacy of chlorpromazine in treating acute psychotic symptoms and reducing relapses was confirmed by a large clinical study making it the first successful APD approved, which was in 1954 [11]. Soon a medley of other APDs followed, termed first-generation APDs, which shared similar modes of action, i.e., the blockade of dopamine D_2_ receptors [8,12]. The first-generation APDs, although efficacious, produced significant side effects limiting their long-term use. Major adverse effects of the first-generation APDs were extrapyramidal symptoms such as acute movement disorders (e.g., dystonia, akathisia, tardive dyskinesia, etc.), parkinsonian-like symptoms, and anticholinergic side effects. In addition, the first-generation APDs were found to negatively affect some aspects of the disease such as weakening knowledge acquisition and cognition, and augmenting hostile behaviours, aggression, and suicidal tendencies [13,14,15]. It has also been reported that some dopamine blocking drugs themselves cause secondary negative and cognitive symptoms in schizophrenia patients [16,17]. Additionally, recurrence of psychotic symptoms has been reported after the withdrawal of some antipsychotic medications. In schizophrenia patients, the quality of life of the patient is greatly compromised due to the long-term exposure to such adverse effects of antipsychotic medications predisposing them to various illnesses such as cardiovascular, metabolic complications, sexual dysfunction, and heightening suicidal tendencies, which increases mortality [18,19,20,21,22,23,24]. The efficacy of all APDs is primarily associated with their action on post-synaptic dopamine D_2_ receptors thereby preventing dopamine hyperactivity in the striatum. However, schizophrenia symptoms that are linked to low dopamine functioning in the prefrontal cortex and certain subcortical areas could not be treated adequately by the first-generation APDs. Additionally, in 2012, Demjaha et al. reported that antipsychotic medications did not elevate the dopamine synthesising capacity of dopaminergic neurons linked to the psychotic symptoms [25].

Antipsychotic medication, for example, aripiprazole and others that were developed later were found to block dopamine D_2_ receptors as well as serotonin receptors. Aripiprazole was reported as a partial agonist at dopamine D_2_ receptors and as an antagonist at serotonin (5-hydroxytrptamine; 5-HT) 5-HT_1A_ and 5-HT_2A_ receptors. They were referred to as the second-generation APDs. The most commonly involved receptor target for the second-generation APDs, beside the dopamine receptors, was found to be the serotonin 5-HT_2A_ receptors [26]. In the second-generation APDs, clozapine was the first APD with superior antipsychotic activities. It differed from the older, classical APDs in superiority of controlling positive as well as negative symptoms, preventing relapses, and also inducing negligible extrapyramidal side-effects [27,28,29]. The mode of action of clozapine was termed atypical because of its actions on multiple receptor sites, predominantly the dopamine D2 receptor blockade and also affinity for 5-HT, acetylcholine, and histamine receptors [20]. Clozapine binds strongly to dopamine D4 and serotonin 5-HT_2A_ receptors and displays weak binding with dopamine D_1_, D_2_, and D_3_ receptors. It has been reported that clozapine causes lesser risk of inducing extra-pyramidal symptoms, increasing prolactin levels and the induction of tardive dyskinesia upon long-term use. These actions of clozapine were associated with its weak binding to dopamine D_2_ receptors and strong affinity to serotonin 5-HT_2A_ receptors in the striatum [30,31]. The second-generation APDs, beside clozapine, included drugs such as amisulpride, aripiprazole, asenapine, brexpiprazole, cariprazine, iloperidone, lurasidone, olanzapine, paliperidone, quetiapine, risperidone, sertindole, ziprasidone, and zotepine. These drugs effectively managed schizophrenia patients who were unresponsive to the first-generation APDs [27]. Improvement in several schizophrenia indices such as anhedonia, avolition, reduced motivation and self-drive, learning and performance tests, depression, adherence, time to relapse, aggression, and suicidal tendencies were successfully achieved with the current APDs [32,33,34,35,36,37,38]. Schizophrenic patients require chronic or sometimes lifelong treatment with APDs, which predisposes the adverse consequences of APDs therapy. However, the use of second-generation ADPs demonstrated significant improvement in long term adverse consequences compared to first generation drugs. Despite the superiority of second-generation APDs, they were also reported to produce some harmful effects that negatively affected the therapeutic outcomes in schizophrenia.

These negative effects of second-generation APDs included the elevation of cardiometabolic abnormalities (for example, resistance to insulin, glucose intolerance, dyslipidaemia, raised blood pressure, and obesity), among others include life-threatening disorders such as myocarditis, agranulocytosis, and complications such as increased prolactin levels [39]. Additionally, the majority of studies suggested that the currently approved APDs mostly targeted schizophrenia symptoms as the primary intention to treat, therapeutically, and were largely similar in terms of efficacy [28,29,40,41,42]. Thus, the currently available APDs succeeded in lowering most of the aspects of schizophrenia pathology but functional outcomes in an appreciable number of patients have remained poor. Hence, even the newer APDs have somehow failed to achieve the desired therapeutic outcome. Additionally, sometimes the newer APDs were no better than the older APDs because many patients continue to suffer from schizophrenia symptoms and cognitive debilities. Despite the rapid advances in understanding the neuroscience of such neuropsychiatric illnesses, there remains a dearth of drug development activity in schizophrenia. The future APD development programs need to address the unmet goals and challenges of the present schizophrenia pharmacotherapy. The very first challenge in this context would be to find treatments for patients who are nonresponsive to the current schizophrenia medications [43,44,45]. Secondly, a disease modifying drug is the need of the hour, which could control subcortical dopamine levels and have improved outcomes on the course of schizophrenia [45]. In this regard, the recent development of a novel trace amine-associated receptor 1 target including the development of SEP-363856, a new non-dopamine D_2_-receptor binding antipsychotic agent, suggest that meaningful therapeutic advances are taking place and are being pursued with more complex neurodevelopmental and molecular models [46,47,48]. Another recent development in this context is the therapeutic exploration of psychedelic compounds in a variety of neuropsychiatric disorders. Individualised treatment with psychedelic compounds has been suggested based on efficacy and tolerability of such medications. The clinical use of psychedelics is being seen as a paradigm shift in the treatment of neuropsychiatric diseases. A recent systematic review of studies of patients who were treated with serotonergic psychedelic drugs reported early evidence that clinical psychedelics were efficacious and well tolerated in a range of psychiatric conditions [49] leading to an increase in advocacy for the controlled, clinical use of psychedelics including LSD and psilocybin [50]. Hence, psychedelics could be an excellent scientific tool for the development of novel drug molecules for a wide range of intractable neuropsychiatric disorders including schizophrenia, and particularly drug-resistant schizophrenia, and may constitute an exciting new treatment avenue in a health area with major unmet needs. Additionally, exploration of psychedelics would facilitate understanding of the serotonin–glutamate receptor hypotheses, as the serotonin–glutamate receptor complex is the target for both the psychedelic drug compounds as well as the atypical and glutamate classes of APDs [51,52]. A renaissance in the interest to study the effects of psychedelics in the treatment of psychiatric disorders warrants a better understanding of the neurobiological mechanisms underlying the effects of psychedelics in schizophrenia [53]. This narrative review is an attempt to delve deep into neurobiological and neuropsychiatry mechanisms of psychedelics modelling vis-à-vis their role in future drug development in neuropsychiatric disorders including schizophrenia.

### 1.2. Psychedelic Drugs in Schizophrenia

Psychedelics refer to a class of drugs that have hallucinogenic actions on the human brain and that, according to Jaffe, are defined as drug molecules with the ability to cause strong changes in perception, thought, and feeling in human beings, which otherwise are not felt normally except in sleep or at times of religious invocations [54]. Previously, it was reported that serotonin agonists in the human brain behave as hallucinogens as they have a powerful influence on memory, learning, perception and emotion, causing momentary symptoms of psychosis [55]. The pharmacological actions of these compounds have been primarily attributed to their binding with and activation of serotonin 5-HT_2A_ receptor subtypes in the brain. Psychedelic drugs have been reported to produce effects at the cellular and molecular levels in the brain, which probably explains their potential use in a number of etiologically varied psychiatric illnesses (Figure 1) [56]. The administration of low, sub-hallucinogenic doses of psychedelics on a chronic, intermittent schedule is referred to as psychedelic microdosing, which is becoming increasingly prevalent among youths as it is believed to lower depression and anxiety and also improve cognitive function and promote social interaction [57,58,59]. In clinical studies, psychedelic compounds have been found to be well-tolerated and efficacious in regulated doses suggesting that they could open the gate to novel therapeutic approaches for treating various neurological illnesses including schizophrenia. Previous studies have reported that psychedelic compounds could benefit patients suffering from various neurological disorders such as anxiety and resistant depression, substance use disorders, posttraumatic stress disorder, alcoholism and schizophrenia [55,60,61,62]. Additionally, psychedelics have been found to promote de-addiction to tobacco and alcohol, and treat some inflammatory conditions [61,62]. According to recent reports, medically supervised doses of psychedelic drugs are well-tolerated and there was no link of psychedelic drug use to adverse mental-health problems. On the contrary, people who used them reported fewer suicidal thoughts and felt better in several indices of anxiety and depression, which precedes the occurrence of schizophrenia [61,63]. A recent study by Hibicke et al., revealed that psychedelic compounds lowered depressive symptoms in a rodent model of depression suggesting that the therapeutic potential of classic psychedelics might be better than ketamine [64].

In 2019, Cameron et al., in a preclinical study involving male rats reported that small doses of *N*, *N*-dimethyltryptamine (DMT) administered intermittently for a long period of time reduced depressive symptoms and allayed fear of learning without affecting other behaviours. Additionally, in an experiment on male rats, DMT fed rats became fat and healthy during the study period indicating that psychedelic microdosing could improve symptoms of mood and anxiety disorders, although the potential hazards of this finding are still being investigated [58]. Recent studies by Cameron et al., have reported that psychedelic microdosing is relatively common and is subjectively associated with a broad spectrum of socio-affective, cognitive, and physical outcomes [57,59]. Another recent study reported that the use of repeated, low doses of LSD improved mood or cognition [65]. It was observed that low doses of LSD produced negligible subjective changes, altered brain connectivity in limbic circuits. However, it is yet to be examined whether neural changes could be attributed to the drug’s purported antidepressant effect [65]. Further, Raval et al., reported that the antidepressant effects of psilocybin were linked to increased formation of neuronal synapses and reduced expression of the serotonin 5-HT_2A_ receptor [66]. The finding from a recent phase 2, double-blind, randomized, controlled clinical study involving patients with moderate-to-severe major depressive disorder reported that psilocybin exhibited superior efficacy over escitalopram [67]. Another recent clinical study suggested that psychedelic compounds including ketamine could be developed as fast-acting antidepressants and their use in psychiatry would represent a paradigm shift in approaches to treating brain disorders by focusing less on rectifying “chemical imbalances” and placing greater emphasis on achieving selective modulation of neural circuits [68]. Research into psychedelic compounds is believed to hold promise in realising antipsychotic medication that could cure neuropsychiatric disorders by correcting the underlying pathophysiology instead of merely treating the disease symptoms [69]. Psychedelic-assisted pharmacotherapy has been reported to have positive therapeutic outcomes in the treatment of trauma, childhood assault and injuries, poor economic and financial conditions, abuse, and deprivation and other associated mental afflictions that usually have adverse effects on mental health [70]. Further, recent clinical studies have provided evidence to support the therapeutic benefits of lysergic acid diethylamide (LSD), 3,4-methylenedioxymethamphetamine (MDMA), 2,5-dimethoxy-4-iodoamphetamine (DOI), psilocybin, and ayahuasca in treating mood and anxiety disorders, severe depression, trauma and stress-related disorders, post-traumatic stress disorder (PTSD), and substance use disorders as well as in end-of-life care [71]. Recently, psychedelic drugs including psilocybin and 3,4-methylenedioxy methamphetamine (MDMA) have been given the status of a breakthrough medicine by the FDA in the United States [71]. According to a recent review by Tahseen Noorani, pharmaceuticalization of psychedelic therapy is gaining momentum and the development of psychedelic-assisted treatments definitely hold ample opportunities [72].

Psychedelic compounds have been grouped into two broad categories: tryptamines and phenethylamines [73,74]. The tryptamines chemically resemble the neurotransmitter and tissue hormone serotonin [73], and the psychedelic compounds (Table 1) included in this group are psilocybin (magic mushroom), lysergic acid diethylamide (LSD), and DMT. The psychedelic compounds belonging to the latter category include mescaline. *N*, *N*-diethyltryptamine (DET) and *N*, *N*-dipropyltryptamine (DPT) are newer analogues of DMT.

### 1.3. Psilocybin

Psilocybin (4-phosphoryloxy-*N*, *N* dimethyltryptamine) is a widely known contraband substance and occurs naturally as an indole alkaloid. In 2015, Hendricks and colleagues reported that psilocybin showed positive effects on mental health such as reduced psychological distress and suicidal feelings [63]. Preliminary findings of phase II clinical studies have suggested efficacy of psilocybin in conditions including obsessive compulsive disorder (OCD), depressive disorder, cancer-induced anxiety, and substance use disorders such as alcohol and tobacco [55]. In 2006, a study by Moreno and colleagues found that there was a greater reduction in OCD symptom following one or more sessions with psilocybin, which was administered at doses ranging from 25, 100, 200, and 300 µg/kg and given at an interval of 1 week. Additionally, the study patients mostly reported that they experienced relief even after psilocybin has left the body, beyond the 24 h assessment [75]. In the amygdala of the human brain, psilocybin has been reported to possess an inhibitory effect, which may explain the observed positive affective state with psilocybin use. In patients of drug resistant depression, a recent phase II clinical study reported that Hamilton Depression Rating Scale scores improved following treatment with psilocybin [76] indicating therapeutic efficacy in depression. Patients who have suffered from schizophrenia for a long time tend to develop depression, anxiety and substance use disorders, and develop other secondary illnesses that become challenging to treat besides the psychotic symptoms. Additionally, according to a recent review, psilocybin lowered suicidal tendencies, which are common in psychotic patients. Psilocybin has been reported to modulate the thoughts and behavioural patterns in individuals who are at risk of suicidal behaviours. The mode of action of psilocybin is suggested to be the regulation of major pathways linked with suicidal behaviours which are affected by directly activating serotonin 5HT_2A_ receptors, and also, targeting the inflammatory and oxidative stress pathways leading to neuronal plasticity, and suppression of inflammation and increase in cognitive flexibility [77]. Recently, the FDA has dubbed psilocybin as a breakthrough medicine for the treatment of resistant depression [78]. Hence, future clinical studies involving psilocybin hold promise in the development of a novel drug for schizophrenia particularly for patients who develop suicidal behaviours and suicidal ideations.

### 1.4. LSD

LSD (lysergic acid diethylamide) is a semisynthetic ergot alkaloid [73,79] that was developed as a means of mind control by security agencies [80,81]. Its therapeutic dose range has been suggested to be between 100 and 200 μg, although it exhibits psychoactive effects at doses as low as 20 μg. Before the ban on psychedelics was imposed through a 1971 United Nation convention [82], many studies had already reported the benefits of LSD in several neurobehavioral conditions such as substance use disorder, pain, neurosis, and cancer-related anxiety, depression, and mood disorders, among others [83,84,85,86]. In a recent study, a 20 µg dose of LSD significantly improved tolerance time to cold (3 °C) water and reduced experience of pain and unpleasantness, subjectively [86] Additionally, a recent study reported that the administration of a 200 μg dose of LSD increased emotional empathy, which was attributed to an increase in oxytocin levels [87]. A clinical study reported a reduction in anxiety and increased quality of life following treatment with LSD for 12 months at a dose of 200 μg, and it was also found to be well-tolerated [88,89]. Further, a recent study suggested that oral administration of 5 μg, 10 μg, and 20 μg of LSD every fourth day over a 21-day period was safe and tolerated, and it is being explored for the treatment and prevention of Alzheimer’s disease [90]. Another recent pilot study reported that the intake of LSD led to increased positive thinking in healthy human subjects [91], and augmented emotional responses to music [92].

The mechanism through which LSD mediates the majority of its pharmacological actions has been primarily attributed to the activation of serotonin 5HT_2A_ receptors, but it also has some modulatory effects on 5HT_2C_ and 5HT_1A_ receptors. However, these receptor interactions and activation, which results in cognitive impairment and hallucinations, and the details of these receptors are still being uncovered. The activation of the 5-HT_2A_ receptor by LSD has been reported to cause the breakdown of inhibitory processes in the hippocampus of the prefrontal cortex. This leads to a reduction in brain neural activity in key areas of the brain. Additionally, activation of the right hemisphere, alterations in thalamic functions, and increases in neuronal activities in paralimbic and frontal cortical structures have been suggested to form visual imageries [79,93]. Further, LSD has been shown to affect the expression of brain-derived neurotrophic factor (BDNF) and glial cell line-derived neurotrophic factor. These are proteins that play key roles in neuronal growth, neuroplasticity, learning, and memory [79,93], and also induce remodelling of pyramidal neurons. A clinical study reported that increases in primary process thinking induced by LSD was mediated by the activation of 5-HT_2A_ receptors leading to disembodiment and a blissful state [94]. Additionally, a study reported that LSD reduced associative, and at the same time increased sensory-somatomotor brain-wide and thalamic connectivity, which was blocked by ketanserin, indicating that LSD’s psychedelic effects were due to activation of 5-HT_2A_ receptors [95]. A recent study reported that the administration of LSD could have anti-inflammatory properties, which was also suggested to be via 5-HT_2A_ receptor signalling [90].

Suggestibility refers to the state of an individual where the subject is inclined (and willing to accept) the actions or suggestions of others. Previously, suggestibility was linked to enhanced therapeutic efficacy of LSD in various diseased conditions including pain [96] and depressive disorders [97]. LSD mediated enhancement of suggestibility and creative imagination could be correlated with therapeutic benefits in psychiatric conditions including schizophrenia. The findings from several studies have rekindled the therapeutic interest in research involving LSD in a range of neuropsychiatric disorders including anxiety, depression, pain, posttraumatic stress disorders, and psychotic conditions [98].

### 1.5. MDMA

3,4-methylenedioxymethamphetamine (MDMA) was first synthesised in 1912. It is a derivative of methamphetamine and used as a middle agent in the synthesis of other chemical compounds [99]. It gained popularity as an ‘Ecstasy medicine’ before being banned as a controlled substance. MDMA exhibits pharmacological actions of both methamphetamine and mescaline. It is a strong releaser of catecholamine neurotransmitters through action at presynaptic reuptake sites, similar to the actions of methamphetamine. In addition, it has strong pre-synaptic serotonin releasing activities. 5-HT_2A_ receptor antagonists such as ketanserin have been found to diminish the subjective effects of MDMA similar to that observed with mescaline and other classic hallucinogens [100]. A recent randomized clinical trial (RCT) has shown efficacy of MDMA-assisted psychotherapy in severe post-traumatic stress disorder [101]. The findings of another RCT reported that MDMA-assisted therapy was highly efficacious and tolerable in patients with severe PTSD, including common comorbid neuropsychiatric disorders such as dissociation, depression, a history of alcohol and substance use disorders, and childhood trauma [102].

### 1.6. DMT

DMT possesses a similar molecular composition and affinity for binding to 5-HT_2A_ receptors as psilocybin and LSD, but exhibits several different pharmacological actions [55,103,104]. DMT also binds to 5-HT_2C_ and 5-HT_1A_ receptors, which confers it with a variety of pharmacological actions. Some pharmacological actions of DMT have been linked to its binding to and activation of sigma-1 and trace-amine receptors, among others [105,106]. In 1965, Franzen and his colleague found the presence of DMT in the biological fluids of healthy individuals [107]. In 2012, Barker and colleagues found evidence from many studies that indicated the presence of DMT in the biological samples of both schizophrenia patients and control subjects who had never consumed it [108]. In animals, DMT was found to be present in the brain and pineal gland of rodents [109,110,111]. In 1999, Thompson and colleagues reported that indolethylamine-*N*-methyltransferase (INMT), the enzyme that produces DMT from tryptamine, has a ubiquitous presence in human organs such as lungs, thyroid, adrenal glands, placenta, skeletal muscle, heart, small intestine, stomach, pancreas, and lymph nodes [112]. According to a published report, DMT is responsible for neurocognitive activities such as consciousness and perception, particularly visual perception. It mediates neurocognitive activities through trace amine associated receptors [113]. Previous studies involving closely related synthetic DMT analogue have indicated that DMT could be used as an adjunctive psychotherapy for the treatment of alcohol addicts [114,115,116] and cancer diagnosis-induced anxiety [115,117,118,119]. DMT has also been found to modulate immune function and reduce inflammation by activation of sigma-1 receptor-mediated pathways [106,120]. In 2007, Heekeren and colleagues reported that DMT diminished the magnitude of the startle response in schizophrenia patients but not in healthy individuals [121]. The study of DMT has helped in understanding the differences between psychosis caused by hallucinogenic compounds and those that occur naturally, and has also increased our understanding of organically occurring psychotic disorders and exhibited symptoms [122]. In recent studies, DMT has been found to be well-tolerated, providing ample opportunities to investigate the potential of DMT. In a recent review, it was proposed that DMT and other psychedelics could play substantial roles in the development, growth, maintenance, and repair of the brain [123].

### 1.7. Mescaline

In 1896, Arthur Heffter isolated mescaline (3,4,5-trimethoxy-β-phenethylamine), which occurs in nature as a psychedelic compound, from *Lophophora williamsii* [124]. The pharmacological actions and adverse effects of mescaline have been reported to be comparable to LSD and psilocybin [125,126]. An early observational study found mescaline to be well-tolerated and efficacious in substance use disorders, e.g., alcohol addiction [127], and recent data supported these findings. It was found that long-term users of mescaline elicited no impairment of cognition in comparison to the drug naive controls. In addition, the use of mescaline was found to cause significantly greater psychological well-being and general positive behaviours in comparison to controls [128].

## 2. Psychedelic Drug Models in Schizophrenia

Animal models of schizophrenia have been developed based on the observation that psychotomimetic drugs induce similar behavioural responses to schizophrenia in humans and animals. Phencyclidine (PCP)-like drugs such as MK-801 and ketamine are uncompetitive blockers of *N*-methyl-d-aspartate (NMDA) receptors, and psychedelic drugs such as LSD, mescaline, and psilocybin are activators of 5HT_2A_ receptor subtypes. PCP and LSD cause greater alterations in sensory perception, and the intake of these drugs produce significant distortions in the perception of environmental stimuli and develop self-generated hallucinations [73]. Other psychoactive effects of these drugs include significant distortion of sensory processing, cognitive functioning, alteration in brain metabolism and distortion of self-image. Ketamine and MK801 are pharmacologically similar to PCP, eliciting strong positive and negative schizophrenic symptoms and impairing cognition, mimicking symptoms of schizophrenia. These agents are widely used to produce schizophrenia symptoms in experimental animals [129,130]. PCP is used to model negative symptoms together with catatonic features of schizophrenia, while LSD could be used to model paranoid types of schizophrenia. Psychedelics are reported to alter visual perception and mimic early schizophrenia symptoms, which mostly include visual disturbances. However, schizophrenia is frequently associated with auditory hallucinations. Research is underway to identify the underlying neurobiological mechanisms in animal models, which could provide insights into possible related mechanisms underlying psychosis [51]. However, the identification of potential biomarkers for neuropsychiatric disorders is of paramount importance for understanding how psychedelics bring about changes in such biomarkers, and for a deeper understanding of their mechanisms of action. Hence, in the last few years, a great deal of effort has been made to identify biomarkers that have the potential to improve prevention and diagnosis, and to help study the drug response for developing new drugs for psychiatric disorders. These biomarkers have been divided based on their main clinical application: diagnostics, monitoring, pharmacodynamic/response, predictive/prognostics, safety, and susceptibility/risk biomarkers [131].

### Biomarkers of Neuropsychiatry and Their Association with Psychedelics Drugs

Biomarkers in neuropsychiatry are becoming immensely important and are playing a key role in facilitating diagnosis of the disorders, and the specific targeting of such biomarkers is helping in targeted treatments [131]. The findings of brain imaging studies have demonstrated an increase in the activation of neuronal structures involved in attention, reward perception, action selection, decision making and behaviour control following response to a drug therapy [132] in several brain regions [133,134,135], and altered neurochemicals in these brain areas was linked to drug craving [136]. Dopamine is a key neurotransmitter involved in various processes associated with cognition such as execution, decision-making, and planning, and also reinforces actions associated with reward and positive thinking [136]. Repetitive consumption of a drug leads to an increase in dopaminergic firing resulting in a rise in dopamine levels in brain areas such the anterior cingulate cortex, amygdala and nucleus accumbens [137,138]. Dopamine is also released along with glutamate in brain areas such as the nucleus accumbens, ventral tegmental area and prefrontal cortex associated with impulsivity, attentional, motivational and emotional processes following stimulus by addictive drugs. 5-HT is considered to be the regulator of emotion, stress and appetite, and is found to be increased in substance use disorders. Additionally, various neuropsychiatric symptoms such as anhedonia, dysphoria, depression, and anxiety during abstinence have been associated with altered metabolism of serotonin leading to triggering of drug seeking behaviours in human [138,139,140]. Further, in neuropsychiatric disorders, access to brain samples is particularly valuable; however, systematic investigations involving brain samples are limited because of the difficultly in monitoring the course of the disease. Functional neuroimaging techniques have been used for studying neuronal activities, alterations in local cerebral flow, energy metabolism and neurotransmitter receptor populations and function during the course of disease. However, they have failed to provide insights at the cellular biochemistry level and are limited due to their high economic costs. In this regard, in recent years, blood lymphocytes are increasingly being used as peripheral biomarkers for studying various diseases including neurological and psychiatric disorders because of the ease in sampling and isolation, and they also allow for daily monitoring of disease course [141]. It has been observed that the lymphocyte-mediated release of cytokines affects neuroendocrine and neurobehavioral responses including autonomic control. Furthermore, it was found that disruption of lymphocyte functions and metabolism leads to changes in neurotransmitters and the hypothalamic–pituitary–adrenal axis [142]. Hence, it was suggested that studies on lymphocyte gene expression in psychiatric patients who are at different stages of the disease could forecast alterations in neuronal activities in the brain. Additionally, this would help in characterising the mechanisms underlying the pathogenesis of the disease and in predicting the outcome of the pharmacological treatment. Lymphocytes obtained from the blood of psychiatric patients, such as schizophrenic and depressive patients, have been analysed for some proteins including c-fos, interleukins (IL) such as IL-2, IL-4, IL-6, and IL-10; nerve growth factor (NGF); and BDNF. Additionally, cannabinoid receptors, cholinergic receptors, γ-aminobutyric acid-A(GABA_A_) receptors, β_2_ adrenergic receptors, glucocorticoid receptors, mineralocorticoid receptors, dopamine D_3_ receptor, and 5-HT receptors have been analysed in lymphocytes from patients of schizophrenia and depression, suggesting that lymphocytes are important peripheral biomarkers [34,35,36,37,38,39,40,41].

Genomic biomarkers have also been used to study normal biologic and/or pathogenic processes, including pharmacotherapeutic responses. These biomarkers are measurable features of deoxyribonucleic acid (DNA) and/or ribonucleic acid (RNA), such as single nucleotide polymorphisms (SNPs), variability of short sequence repeats, haplotypes, deletions or insertions of (a) single nucleotide (s), copy number variations and cytogenetic rearrangements (translocations, duplications, deletions, or inversions) (45). The use of genetic techniques allowed the analysis of candidate genes, genome-wide studies and polygenetic risk score analysis to understand multiple psychiatric disorders including schizophrenia (46, 47). In 2019, Wu et al., reported that an SNP in the gene expressing a protein known as glutamate decarboxylase-like protein-1, was linked with the response to lithium in Chinese bipolar disorder patients [143]. An SNP has been also been associated with immune disturbances in bipolar disease patients, leading to increases in the levels of total T cells, CD4+ T cells, activated B cells and monocytes. Therefore, alterations in the number of immune cells may serve as a biomarker for diagnosis, disease progression, and response to therapy in patients with bipolar disorder. Another class of biomarker in neuropsychiatry are the transcriptomic biomarkers, which have potential for understanding the biology of psychiatric disorders. A transcriptome is the full range of messenger RNA, or mRNA, molecules expressed by an organism. The term “transcriptome” can also be used to describe the array of mRNA transcripts produced in a particular cell or tissue type, alternatively it is a complete set of all RNA molecules present in a single cell or in a cell population at a particular developmental stage or physiological condition [144]. The findings of transcriptomics studies have suggested that the therapeutic responses to antidepressants is linked to changes in the expression of certain genes such as matrix metallopeptidase 28 and K × DL motif-containing protein-1. In a study, Hodgkin et al. observed that efficacy of nortriptyline in patients of depression was linked to changes in genes responsible for synthesising these proteins [145]. The findings of studies on RNA have led to the identification of biomarkers of suicide. The postmortem analysis of the brain areas, particularly the dorsolateral prefrontal cortex and the anterior cingulate cortex (ACC), of depressive patients who committed suicide found altered RNA editing on the cyclic nucleotide phosphodiesterase (PDE), particularly PDE8A, involved in the hydroxylation of cyclic adenosine monophosphate and cyclic guanosine monophosphate. These alterations have been proposed to be a potential biomarker of risk for attempting suicide in patients with depressive symptoms (68). MicroRNA-124 (miR-124) and microRNA-181 (miR-181) were found to be upregulated in the blood samples of females with cocaine addiction, and have been proposed as potential biomarkers for cocaine use disorder [146].

Proteomics is another valuable technique for identifying potential biomarkers for psychiatric disorders. This technique commonly uses blood, plasma or serum including cerebrospinal fluid (CSF) as biological samples for diagnostic purposes in clinical practice, and are easy to obtain [147]. In 2015, Nascimento and Daniel Martins-de-Souza reported that proteomics could help understand the biochemical processes of schizophrenia at the cellular and tissue levels by identifying proteins expressed predominantly in the brain tissue [148]. In 2018, Comes at al., identified alterations in specific proteins in patients with schizophrenia using proteomic studies, and suggested that these proteins could act as potential biomarkers for schizophrenia as they play key roles in relevant pathophysiological, biochemical and neurochemical processes [148,149]. In 2018, Xu et al., reported that one of these proteins was zinc finger protein 729. They found that the expression of zinc finger protein 729 was significantly lower in psychiatric patients in comparison to healthy people [150]. In 2019, Rodrigues-Amorim et al., reported that the levels of specific proteins including glia maturation factor beta, BDNF, and Rab3 GTPase activating protein catalytic subunit (RAB3GAP1) were significantly reduced in the plasma of schizophrenia patients. These proteins were presented as promising biomarkers for this psychiatric disorder [151]. Other potential biomarkers reported for certain psychiatric and neurodegenerative disorders including major depression, anorexia nervosa, bipolar disorders, etc., have been found to be acetyl-L-carnitine and neurofilaments light chains [152,153,154]. Lately, a number of studies have identified potential metabolomics biomarkers in different psychiatric diseases including schizophrenia and bipolar disorder [155,156], suggesting that metabolomics could be a promising tool for developing precision medicine in psychiatry [156].

Epigenetic biomarkers are another recent development in psychiatry. These are defined as “any epigenetic mark or altered epigenetic mechanism that can be measured in the body fluids or tissues and that could help in the detection of a disease; prediction of the disease outcome; have predictive value in the response to a therapy or medication and could also help in therapy monitoring; and at the same time predict the risk of disease development in future” [157]. Recent studies have found significant correlations of psychiatric diseases with epigenetic modulations of genes regulating neurotransmission, neurodevelopment, and immune function [158,159]. Examples of epigenetic modulations of genes include hypermethylation of genes such as BDNF and FKBP5 found in the brain and blood samples of psychiatric patients suffering from schizophrenia, major depression, bipolar disorder, and post-traumatic stress disorders [160,161,162]. Additionally, hypomethylation of some genes such as monoamine oxidase A and glutamate decarboxylases 1 have been reported [163]. Epigenetic mechanisms, including DNA methylation, miRNA, and histone modifications have been able to explain the complex interplay of environmental risk factors with genetic risk factors in the emergence of suicidal behaviour. In a recent study, epigenetic alterations in key elements of the hypothalamus-pituitary-adrenal axis, brain derived neurotrophic factors, serotoninergic and GABAergic systems were suggested to act as epigenetic biomarkers for suicidal behaviours [164]. Conventional antidepressants targeting monoaminergic neurotransmission has been found to modulate epigenetic mechanisms, indicating epigenetic dysregulation in the pathophysiology of depression [165,166,167]. Recent studies have found that pharmacological regulation of histone acetylation could be a promising clinical strategy in major depression treatment. Hence, histone acetylation status has been proposed as a potential diagnostic biomarker for major depression. Recent studies have shown that inhibition of histone deacetylases (HDAC) increased the expression of BDNF, leading to enhanced neural/synaptic plasticity, and exerting an antidepressant-like effect on behaviour [165,166]. Therefore, HDACs have been proposed as potential diagnostic and therapeutic targets for depression [165].

## 3. Neurochemical Mechanisms of Psychedelic Drugs and Their Relationship with APDs 

### 3.1. Psychedelic Pharmacodynamics and Receptor Activation

The early established drugs for schizophrenia are known to work by blocking dopamine D_2_ receptors, and amphetamine-like drugs increase dopamine concentration at synapses and exacerbate psychotic behaviours. A potent dopamine D_2_ receptor blocker, eticlopride, lacks antipsychotic properties and has been shown to increase dopamine D_2_ receptors expression and activities. Further, it was observed that hyperactivity of dopamine D_2_ receptor neurotransmission in subcortical and limbic brain regions contributed to the positive symptoms of schizophrenia, whereas negative and cognitive symptoms of the disorder were attributed to the dopamine D_1_ receptor-mediated neurotransmission hypofunction in the prefrontal cortex [168,169]. In schizophrenia, neurotransmitters such as 5HT, glutamate, and dopamine play important roles in the pathophysiology of schizophrenia. The mode of action of atypical APDs involves primarily the antagonism of serotonin 5-HT_2A_ receptors rather than the antagonism of dopamine D_2_ receptors. Similarly, the involvement of glutamate was suggested due to the fact that NMDA receptor blockers elicit a schizophrenia-like state. Classic psychedelics including psilocybin, DMT, and LSD have been reported as partial agonists at various serotonergic (5-HT_1_, 5-HT_2_, 5-HT_6_ and 5-HT_7_) receptor subtypes, while some psychedelics such as mescaline and DOI were found to selectively activate other serotonergic (5-HT_2A_, 5-HT_2B_ and 5-HT_2C_) receptor subtypes [55,73,170]. Additionally, some psychedelics, e.g., LSD, activate dopaminergic (D_1_ and D_2_ receptors) and adrenergic receptors [73]. Psychedelic-like drugs produce cellular and behavioural responses primarily by binding to 5-HT_2A_ receptors in the cortical neurons, particularly pyramidal neurons, while LSD-like drugs elicit schizophrenia-related symptoms by interacting with 5-HT_2A_ receptors postsynaptically in the cortex. In both animal and human studies, the activation of 5-HT_2A_ receptors in cortical and subcortical structures was associated with the behavioural and psychological effects of psychedelics [171,172,173]. Additionally, in a clinical study, it was demonstrated that subjective effects of psilocybin, LSD and DMT were blocked by ketanserin, a 5-HT_2A_/5-HT_2C_ receptors blocker, and pretreatment with buspirone, a 5-HT_1A_ agonist, lowered the visual hallucination produced by psilocybin [174]. In a separate study, pindolol, a 5-HT_1A_ antagonist, augmented the psychological responses to DMT [175]. Hence, the modulation of 5-HT_1A_ receptor activity on 5-HT_2_ mediated psychedelic effects may be targeted to treat visual hallucinations [174]. Psychedelics have also been reported to produce behavioural and psychological responses through modulation of dopaminergic activity. In a clinical study, euphoria and depersonalization disorders have been linked to an increase in dopamine concentrations in the striatum of the brain by psilocybin [176], and treatment with haloperidol, a dopamine D_2_ receptor blocker, diminished the psilocybin-induced depersonalization in human subjects [171]. Psilocybin was found to indirectly activate dopamine receptors, while LSD directly activated dopamine D_2_ receptors thus, LSD produces greater psychotic-like effects than psilocybin in humans [73]. A recent animal study showed that psychotic-like behavioural effects of LSD was due to the modulation of dopaminergic activity in the ventral tegmental area via activation of trace amine-associated receptors 1 (TAAR1) [177,178]. Therefore, TAAR1 receptors could be novel targets for studying the mechanisms of psychedelics. Hallucinogenic and non-hallucinogenic psychedelic agonists acting via 5-HT_2A_ receptors, such as LSD and lisuride, are distinct in activating intracellular second messenger systems in cortical neurons [179], and it was observed that only psychedelics such as LSD and DOI, which are specifically the hallucinogenic agonists, were capable of increasing the expression of the early growth receptor (EGR)-1 and EGR-2 genes [4]. Such functional dichotomy in the actions of hallucinogenic and non-hallucinogenic agonists are the subject of ongoing investigations.

### 3.2. Agonist Trafficking of Receptor Signalling and Psychedelics

Agonist trafficking of receptor signalling has been suggested as the mechanism of action of hallucinogenic drugs. According to this model, it is suggested that drugs acting as agonists at the same receptor would stabilize active conformational states, which, in turn, would bind with different G-protein subtypes to elicit different cellular signalling responses. This model may be able to explain differing responses elicited by hallucinogenic and non-hallucinogenic drugs acting through the activation of 5-HT_2A_ receptors despite having similarity in their chemical structures and signalling pathways [180,181]. It was proposed that hallucinogenic drugs mimicking LSD and non-hallucinogenic drugs mimicking lisuride-like responses would stabilize 5-HT_2A_ receptor conformations differently, and also modulate different signalling pathways, which are responsible for their unique behavioural effects [51].

### 3.3. Functional Serotoninergic/Glutamatergic Interactions in Psychedelic Responses

Recent studies have indicated that serotonin and glutamate interact functionally, which might clarify some of the unresolved complexity in the mode of action of APDs and is expected to open up new strategies to design new medications for schizophrenia. Glutamate plays a key role in schizophrenia pathophysiology and has been suggested to be the main signalling molecule present in the pyramidal cells, which form the efferent and interconnecting fibres of the cerebral cortex and limbic systems. These areas have been implicated in schizophrenia pathophysiology. A study by Marek and colleagues reported that 5-HT_2A_ and metabotropic glutamate subtypes 2 and 3 (mGlu2/3) receptors were functionally interconnected in the rat prefrontal cortex. The demonstration of a number of cellular, electrophysiological and behavioural effects was explained by the functional interactions of 5-HT_2A_ and mGlu2/3 receptors [182]. The behavioural responses produced by psychedelic drugs are inhibited by the activation of mGlu2/3 receptors, which block the glutamate release from cortical terminals and which are evoked by activity of 5-HT_2A_ receptors present on neurons of thalamic nuclei. Further, a direct interaction of postsynaptic 5-HT_2A_ and mGlu2 receptors has been noted as both the receptors were found on the presynaptic and postsynaptic neurons in the cortex (Figure 2). Psychedelic drugs activate 5HT_2A_ receptors and act via heterotrimeric G proteins, which, in turn, pass the signal through 5-HT_2A_ mGlu2 receptor complexes in cortical neurons. This signalling pathway is inhibited by the agonists of mGlu2 receptors. Hence, it is believed that abnormal functioning of 5-HT_2A_–mGlu2 receptor complexes would result in schizophrenia-like symptoms [4].

### 3.4. Role of mGlu2/3 Receptors in Psychedelic Drug Responses

In 2007, Patil and colleagues reported that the administration of the mGlu2/3 receptor agonist, LY404039, improved psychotic-like symptoms (representing positive and negative symptoms of human schizophrenia) [183] in a metabotropic glutamate receptor knock out mice model, but LY404039, LY314582 and LY379268 were found to mediate their action through mGlu2 receptor activation in an animal model of psychosis. In 2019, Nicolleti and colleagues reported that positive allosteric modulation of various mGlu receptors, selectively, could treat positive symptoms of schizophrenia. There is evidence that mGlu1 receptor activation inhibits the dopamine release in the mesostriatal system, while mGlu5 receptors control the cellular activities observed in schizophrenia. Additionally, mGlu2 receptors were found to negatively modulate 5-HT_2A_ receptor signalling in cortical areas of the brain. In animal models, allosteric modulation of both the mGlu2 and mGlu2/3 receptors exhibited antipsychotic-like activity. Metabotropic glutamatergic, mGlu5, receptors have been found to be widely expressed in the CNS and are under development for the treatment of schizophrenia. So far, mGlu3 receptors have been neglected as drug targets for schizophrenia. However, mGlu3 receptor activation has been reported to facilitate mGlu5 receptor signalling, supporting neuronal survival and forcing microglial cells to exhibit an anti-inflammatory phenotype, encouraging research on mGlu3 receptors in schizophrenia. Finally, preclinical studies have indicated that mGlu4 receptors could be potential targets for novel APDs [184].

There are also studies that have reported functional interactions between 5HT_2A_ and NMDA receptors, besides glutamate receptors with 5HT_2A_ receptors. Antagonism of NMDA receptors produces a variety of behavioural responses in rodents, which include head-twitch and head weaving responses. These behavioural responses have also been recorded in psychedelic drug-treated animals, which was not exhibited by genetically modified mice lacking 5HT_2A_ receptors. Interestingly, however, head-twitch and head-weaving actions were not produced in animals that received *R*-lisuride, *S*-lisuride and ergotamine, non-hallucinogenic, 5HT_2A_ receptor agonists, indicating that head-twitch and head-weaving actions are specific for psychedelic and PCP-like drugs. Nabeshima and colleagues [185] have shown that PCP-like drug-induced head twitch occurred through 5HT_2A_ receptor signalling in rats, and Kim and colleagues [186] reported that the blockade of 5HT_2A_ receptors augmented 5HT_2A_ receptor-mediated behavioural responses including head twitch and head weaving responses. Later on, it was observed that several 5HT_2A_ receptor antagonists including clozapine, M100907, blocked the behavioural responses produced by PCP-like drugs. Subsequently, Kargieman and colleagues reported that APDs reversed the disrupted prefrontal cortex functioning produced by drugs mimicking PCP-like responses [187,188]. A study by Schmid and colleagues suggested that serotonin plays a key role in producing hallucinogenic effects in vivo, as a precursor of serotonin, 5-hydroxytrytophan, produced a similar head twitch response in mice [189]. Drugs mimicking responses of both PCP and LSD utilise signalling via cortical 5HT_2A_ receptors, and further hallucinogen specific signalling produced through the mGlu2 receptor by LSD-like drugs at the 5-HT_2A_–mGlu2 receptor complex was terminated by the activation of mGlu2 receptors (Figure 2). Hallucinogens, newer second-generation antipsychotic medication and metabotropic glutamatergic agonists have been reported to bind to the same 5-HT_2A_–mGlu2 receptor complex. It has been suggested that both the atypical antipsychotic medication and psychedelic compounds have a similar molecular target, which is the 5-HT_2A_ receptor, while the mGlu2 receptor was found to be the molecular target for newer glutamatergic antipsychotic medications.

### 3.5. Role of Central Histamine in Psychedelic Drug Responses

Besides glutamate and serotonin, histamine also has a strong psychoactive connection, which remains incompletely understood. Histamine contributes to the regulation of sleep and wakefulness. It has been reported that during states of wakefulness, central histamine neurons fire rapidly, while this is slow during rest and stops during REM sleep. Blocking of histamine has been found to induce sleep, and histamine release by histamine H_3_ receptor antagonists and or inverse agonists such as ciproxifan and clobenpropit promotes wakefulness and restores cognitive deficits as observed in schizophrenia [190,191]. It is proposed that psychedelic compounds may exhibit positive therapeutic efficacy in neuropsychiatric disorders including schizophrenia through the close interplay of complex neurotransmitter signalling processes along with the involvement of central histamine, 5-HT_2A_, and mGlu2/3 receptors.

### 3.6. Psychedelic Effects in Neural Connectivity and Neural Plasticity

Neural plasticity, which is the ability to change and adapt in response to stimuli, is an essential aspect of healthy brain function and, in principle, can be harnessed to promote recovery from a wide variety of brain disorders. Psychoplastogenic compounds including psychedelic compounds, ketamine, and several others have been recently discovered as fast-acting antidepressant agents. Their use in psychiatry is expected to bring a paradigm shift in our approach to treating brain disorders as we pay less attention to rectifying “chemical imbalances” and more on achieving selective modulation of neural circuits [192]. Psychedelic drugs have been shown to regulate the excitatory-inhibitory balance in neural circuits and participate in neuroplasticity within brain structures that are important for the assimilation of key information relevant to sensation, cognition, emotions, and the narrative of self [53]. In preclinical studies, the administration of LSD and DOI caused an increase in cortical glutamate levels and layer 5 pyramidal cell activity in the prefrontal cortex [193], attributed to repetitive network activity triggered by activation of postsynaptic 5-HT_2A_ receptors located in deep layer 5 or 6 pyramidal neurons that project to layer 5 pyramidal neurons.

The rise in glutamate levels leads to the activation of postsynaptic alpha-amino-3-hydroxy5-methyl-4-isoxazole propionic acid (AMPA) receptors in the same neurons, which was suggested to increase production of BDNF. In a preclinical study on rodent models, treatment with DOI increased the production of BDNF in the prefrontal cortex and hippocampus [194], and the finding from in vitro and in vivo studies reported that treatment with LSD, psilocybin, and DMT including DOI caused both structural and functional neuronal plasticity in prefrontal cortical neurons [195,196]. A recent study demonstrated that low doses of LSD acutely increased levels of BDNF in blood plasma in human subjects [197]. In 2020, Barret and colleagues reported that psilocybin treatment acutely produced effects such as lowering of negative mood, improvement in positive mood, and reduction in amygdala activity in response to negative affective stimuli. It was observed that psilocybin increased emotional and brain plasticity supporting the hypothesis that the negative effects of psilocybin could be a novel therapeutic approach [198]. The ability to selectively modulate neural circuits using small psychedelic molecules could open up new horizons in neuropsychiatry focusing on curing neural circuitry anomalies instead of addressing disease symptoms alone. This type of circuitry approach by psychedelic drugs would lead to a fundamental shift in the treatment approaches to a number of neuropsychiatric diseases and would have great implications, therapeutically, for the future of CNS drug discovery [69]. Additionally, psychedelic drug-induced changes in self-referential processing and emotion regulation have been associated with characteristic changes in brain activity and connectivity patterns at multiple system levels. These alteration in self-experience, emotional processing, and social cognition have been suggested to contribute to the potential therapeutic effects of psychedelics [53].

### 3.7. Limitations and Scope of Psychedelic Studies

Psychedelics are currently prohibited substances in most countries; thus, a growing understanding of their therapeutic potential is leading many people to use psychedelics on their own rather than waiting for legal medical access. The major limitations of psychedelic research are the lack of interest from the pharmaceutical industry and limited support and funding for research have restricted trials to relatively small samples of patients because of the cost of doing larger studies. It is still perceived that if psychedelic drugs are introduced into general clinical practice, there would be a risk of their misuse without sufficient safety and efficacy evidence, in much the same way that happened with medical cannabis [199,200]. Additionally, there is a high chance that individuals might seek underground (i.e., illegal) therapy, travel to countries that have permitted psychedelics use, or use psychedelics on their own [201]. The current study is a narrative literature review that relied largely on the information available in PubMed or Medline databases, involving research on psychedelics and their role in neuropsychiatric drug development. Therefore, despite the potential of psychedelics in psychiatry pharmacotherapy, further work will be needed to ascertain the individualised suitability of psychedelics for diverse mental health conditions, the most appropriate type of psychotherapy to employ, and the psychological and neurobiological mechanisms underlying clinical benefits. This review study is an effort to investigate schizophrenia pathophysiology vis-à-vis the role of psychedelics in modulating common neuronal circuitry to understand the neurochemical mechanisms of psychedelic drugs and their relationship with antipsychotic medication. Additionally, there is a need to delve deeper to understand the modulatory role of psychedelics on biomarkers of neuropsychiatry.

## 4. Conclusions

The dopamine and serotonin hypotheses of schizophrenia continue to hold a vital place in the development of antipsychotic medication. However, the pathophysiology of schizophrenia is still expanding and new theories are emerging. The pharmacological effects produced by psychedelic drugs and some primary schizophrenia symptoms have shown considerably high resemblances. Some atypical APDs and LSD-like drugs act on common sites of action, which are the serotonin 5HT_2A_ receptors. Psychedelic drugs including the newer second generation along with the glutamatergic APDs are thought to mediate pharmacological actions through a common pathway, specifically a complex serotonin–glutamate receptor interaction in cortical neurons of pyramidal origin. Furthermore, psychedelic drugs have been reported to display complex interplays between 5HT_2A_, mGlu2/3 and NMDA receptors for mediation of neurobehavioral and pharmacological actions. The findings from recent studies have suggested that serotoninergic and glutamatergic neurotransmissions are very closely connected in producing pharmacological responses to psychedelic compounds and antipsychotic medication. Emerging hypotheses suggest that psychedelic drugs work through a brain resetting mechanisms and could cure the underlying pathology of neuronal circuitry, which could benefit psychiatric disorders including schizophrenia. Hence, psychedelics currently offer an exciting scientific tool for researchers around the globe to explore new neuropsychiatric approaches for treating a wide range of psychiatric maladies including schizophrenia [123].

## Figures and Tables

**Figure 1 pharmaceuticals-15-00640-f001:**
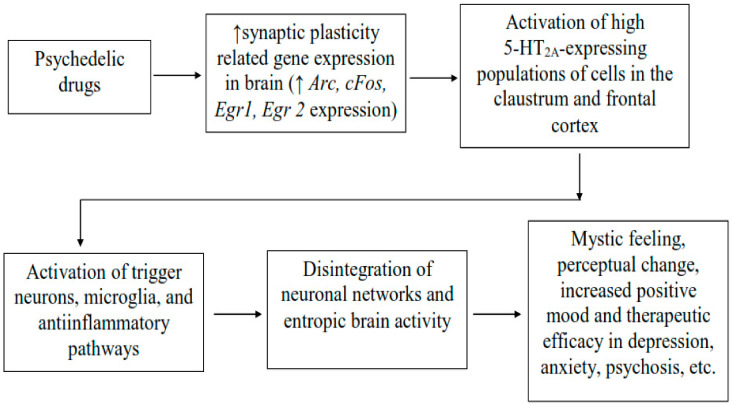
Neurobiological mechanisms of action of psychedelic drugs beside 5HT_2A_ agonism. Psychedelic drugs mediate increase expression of gene relating to synaptic plasticity in the brain initiating downstream neuronal signaling leading to the activation of neurones showing high expression of 5-HT_2A_ receptors in the brain regions such as claustrum and frontal cortex, and which underlie changes in the cortical network and entropic brain activity using psychedelics. The changes in brain activity may lead to the subjective psychedelic experience such as mystic feeling, perceptual changes, increase positive mood and therapeutic efficacy in depression, psychosis, etc. Arc: activity-regulated cytoskeleton-associated protein; Egr1 and Egr2: early growth response protein 1 and 2; cFos: a marker of neuronal activation [56].

**Figure 2 pharmaceuticals-15-00640-f002:**
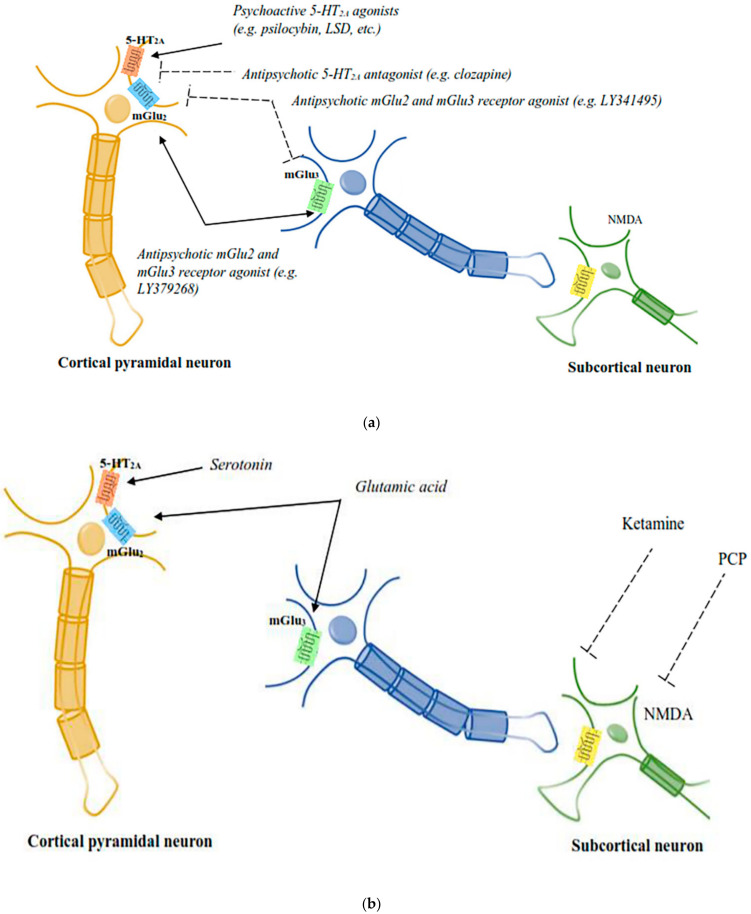
(**a**): The diagrammatic elucidation of the neuronal circuits mediating the responses of psychoactive chemicals and APDs. 5-HT_2A_–mGlu2 receptor complexes expressed by cortical pyramidal neurons are believed to be the site of action for both psychoactive 5-HT_2A_ receptor agonists and mGlu2 receptor antagonists, in addition to antipsychotic 5-HT2A receptor antagonists and mGlu2 receptor agonists [51]. (**b**): The diagrammatic elucidation of the neuronal circuits mediating the responses of psychoactive chemicals and APDs. PCP-like psychoactive drugs activate subcortical NMDA receptors to mediate the release of serotonin and glutamate in the cortex, which is believed to affect the signalling activities of cortical 5-HT_2A_–mGlu2 receptor complexes [51].

**Table 1 pharmaceuticals-15-00640-t001:** Chemical structures of common psychedelics and their resemblance to the neurotransmitter serotonin.

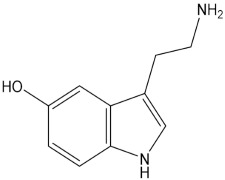	Serotonin
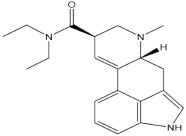	LSD (Lysergic acid diethylamide)
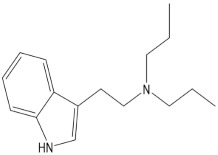	DPT (*N*,*N*-dipropyltryptamine)
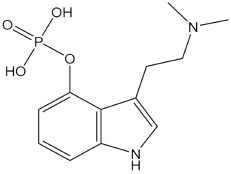	Psilocybin
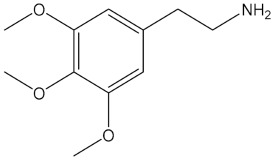	Mescaline
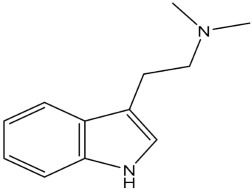	DMT (*N*,*N*-dimethyltryptamine)

## Data Availability

Not applicable.

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
