# Peer review of "New Paradigms of Old Psychedelics in Schizophrenia"

_pharmaceuticals, 2022, doi:10.3390/ph15050640_

Round 1
Reviewer 1 Report
Dear authors,
the manuscript is generally well written and organized.
Please find below some concerns should be addressed before publication according to my experience:
- First paragraph is too long to be considered an “Introduction”. It should be divided into sub-paragraphs with titles.
- Please refer to “Addiction” as “substance use disorder” according to DSM5, otherwise specify which stage of SUD you are referring at.
- About clinical schizophrenia authors should referer to the DSM5 throughout the text (and cite it).
- Positive and negative symptoms deserve to be not only mentioned. please specify extensively.
- Please check carefully any "first time" introducing acronyms and avoid repetition of previously spelled acronyms
- Describing use of drugs as “safe” could sound overly optimistic or naive. I would suggest to substitute with “well tolerated”
- The sentence “This narrative review attempts to highlight psychedelic compounds as the lead in future drug development for a variety of schizophrenia types...” should be toned down. Indeed, despite general hopes, and shareable author’s excitement on the topic, long term experimental research is still needed.
- The sentence “DMT also has been found to modulate immune function and reduce inflammation by activation of sigma-1 receptor mediated pathways” requires precise citation
-The paragraph “NEUROCHEMICAL MECHANISMS OF…” needs more, and precise, citation of studies, which are completely missing in some parts.
- Referring at positive and negative symptoms of schizophrenia in animal models (such as in the sentence “In 2007, Patil and colleagues reported that the administration of mGlu2/3 receptor agonist, LY404039, improved both positive and negative symptoms of schizo-phrenia [111]”) should be strongly limited. Using expressions such as “Psychotic-like simptoms” are suggested for animals.
Author Response
|
Comments |
Responses |
|
Reviewer # 1 Dear authors, The manuscript is generally well written and organized. Please find below some concerns should be addressed before publication according to my experience: |
Thank you very much. |
|
- First paragraph is too long to be considered an “Introduction”. It should be divided into sub-paragraphs with titles. |
As advised, the introduction has been significantly revised with paragraph divided into subparagraphs with titles |
|
- Please refer to “Addiction” as “substance use disorder” according to DSM5, otherwise specify which stage of SUD you are referring at. |
Addiction has been replaced with “substance use disorder” with appropriate citation of DSM-5 |
|
- About clinical schizophrenia authors should referrer to the DSM5 throughout the text (and cite it). |
As suggested, clinical schizophrenia has been referred as per DSM-5 with appropriate citation. |
|
- Positive and negative symptoms deserve to be not only mentioned. Please specify extensively. |
Positive and negative symptoms have been detailed appropriately as per suggestion |
|
- Please check carefully any "first time" introducing acronyms and avoid repetition of previously spelled acronyms |
We have checked all acronyms to avoid repetition of previously spelled acronyms, as suggested. |
|
- Describing use of drugs as “safe” could sound overly optimistic or naive. I would suggest to substitute with “well tolerated” |
The use of term ‘safe’ has been replaced with ‘well tolerated’ |
|
- The sentence “This narrative review attempts to highlight psychedelic compounds as the lead in future drug development for a variety of schizophrenia types...” should be toned down. Indeed, despite general hopes, and shareable author’s excitement on the topic, long term experimental research is still needed. |
The sentence” This narrative review attempts to highlight psychedelic compounds as the lead in future drug development for a variety of schizophrenia types...” has been revised and several new information has been incorporated to pique the excitement of the reader |
|
- The sentence “DMT also has been found to modulate immune function and reduce inflammation by activation of sigma-1 receptor mediated pathways” requires precise citation |
Appropriate citation has been added as suggested |
|
-The paragraph “NEUROCHEMICAL MECHANISMS OF…” needs more, and precise, citation of studies, which are completely missing in some parts. |
More relevant and precise information has been duly incorporated as per the suggestion |
|
- Referring at positive and negative symptoms of schizophrenia in animal models (such as in the sentence “In 2007, Patil and colleagues reported that the administration of mGlu2/3 receptor agonist, LY404039, improved both positive and negative symptoms of schizophrenia [111]”) should be strongly limited. Using expressions such as “Psychotic-like symptoms” are suggested for animals. |
The sentence has been revised as per the suggestion |

Reviewer 2 Report
Dear Editor,
I really appreciate the opportunity to review the manuscript pharmaceuticals-1615113 entitled:
"New paradigms of old psychedelics in schizophrenia"
I commend the authors for describing this critical and timely issue. The paper is interesting and well-written; however, I would like to highlight some issues that merit revision:
1. Limitations are not specified in any point of the text; please, add a short paragraph in the manuscript to describe them
2. From a thorough reading of the paper I could not find any mention of the role of biomarkers in the addiction examined, as well as their role in neuroprogression should be described in more detail. I ask the users to add a short paragraph to discuss these aspects or, alternatively, to add in the limitations why it was not possible to evaluate it.
Author Response
Reviewer # 2
|
Comments |
Responses |
|
Dear Editor, |
Thank you very much. |
|
1. Limitations are not specified in any point of the text; please, add a short paragraph in the manuscript to describe them |
Limitations have been added in the revised version. |
|
2. From a thorough reading of the paper I could not find any mention of the role of biomarkers in the addiction examined, as well as their role in neuroprogression should be described in more detail. I ask the users to add a short paragraph to discuss these aspects or, alternatively, to add in the limitations why it was not possible to evaluate it. |
Role of biomarkers in the addiction as well as their role in neuroprogression has been incorporated appropriately. |

Reviewer 3 Report
The manuscript needs very thorough English language editing as it does not currently meet publication standard.
Organisation needs to be improved throughout, there are many instances of illogical sequencing and repetition of ideas. For instance, in the introduction characteristics of first generation anti-psychotics are discussed, then 2nd generation anti-psychotics are introduced and discussed. But then, more (largely repetitive), details are provided about first generation anti-psychotics.
The introduction needs to be about half current length. Use of subheadings may help organise the text and improve development and flow of ideas.
Anti-psychotic drug sub-sections are a little basic and do not generally provide a deep enough review of material (which also generally needs to be fresher). For instance, the LSD section does not delve deep enough into mechanisms of action and discusses studies that are not fresh (the newest study discussed in this section is now 7-years old). The authors should concentrate on new developments and studies.
Table 2 (which should be Table 1) - what is S. number?. This reviewer doubts that much of the information in this table is accurate. For instance, '15' 'first report of psychodelics/addition - there are many earlier reports than 2014.
Neurochemical mechanisms section could also use subsections to improve organisation and reduce repetition.
The authors should review several other similar published reviews in the area to identify how they may better develop a review that brings something fresh and relevant to readers.
Author Response
We would like to extend our sincere thanks to you for your esteemed comments and suggestions. The manuscript has been revised as per their advice and has been highlighted with color in the manuscript. We believe that addressing all the comments of the reviewers have substantially improved the quality of the manuscript. Detailed point to point replies to the comments are given below.
Reviewer # 3
|
Comments |
Responses |
|
The manuscript needs very thorough English language editing as it does not currently meet publication standard. |
Thanks for the valuable comment. The English language has been improved significantly in the revised version of manuscript. |
|
Organisation needs to be improved throughout, there are many instances of illogical sequencing and repetition of ideas. For instance, in the introduction characteristics of first generation anti-psychotics are discussed, then 2nd generation anti-psychotics are introduced and discussed. But then, more (largely repetitive), details are provided about first generation anti-psychotics. |
Organization of the manuscript has been improved significantly. Illogical sequencing and redundant information has been removed |
|
The introduction needs to be about half current length. Use of subheadings may help organise the text and improve development and flow of ideas. |
The introduction has been divided into subparagraphs with appropriate subheadings/tile |
|
Anti-psychotic drug sub-sections are a little basic and do not generally provide a deep enough review of material (which also generally needs to be fresher). For instance, the LSD section does not delve deep enough into mechanisms of action and discusses studies that are not fresh (the newest study discussed in this section is now 7-years old). The authors should concentrate on new developments and studies. |
Additional fresh information has been added to improve the content |
|
Table 2 (which should be Table 1) - what is S. number. This reviewer doubts that much of the information in this table is accurate. For instance, '15' 'first report of psychodelics/addition - there are many earlier reports than 2014. |
The table has been omitted |
|
Neurochemical mechanisms section could also use subsections to improve organisation and reduce repetition. |
This section has been improved with use of subsection and reduction of |
|
The authors should review several other similar published reviews in the area to identify how they may better develop a review that brings something fresh and relevant to readers. |
Several new articles published in this area has been reviewed and newer and fresh information have been incorporated to significantly improve the manuscript |

Round 2
Reviewer 1 Report
Dear Authors, I appreciated the effort improving the manuscript "New paradigms of old psychedelics in schizophrenia", which is currently well organized and comprehensive.